# Models for Developing Community Organizations to Reinforce Health Management in Small Businesses

**DOI:** 10.3390/ijerph17072444

**Published:** 2020-04-03

**Authors:** Eun-Hi Choi, Hye-Sun Jung

**Affiliations:** 1Department of Nursing, Eulji University, Daejeon 34824, Korea; choieh@eulji.ac.kr; 2Department of Preventive Medicine, The Catholic University of Korea, Seoul 06591, Korea

**Keywords:** community, organizations, small business, empowerment, workplace

## Abstract

The incidence of occupational diseases in small businesses is higher than in big ones, and this fact puts the former in need of a healthcare management model they can administer. This study established a model based on community organizational development theory to reinforce worker empowerment for healthcare in small businesses, focusing on health centers for workers (HCFW). The researchers surveyed 408 workers at 39 small businesses in the B region of South Korea, according to the characteristics of business sites, general characteristics of workers, and elements of community organizational development theory, and analyzed their results with a structural equation. The research period was September, 2015. Five concepts were examined: empowerment in healthcare, community capacity, participation and relevance, critical consciousness, and issue selection. The results revealed that greater community capacity led to greater participation and relevance (γ = 0.39) and empowerment in healthcare (γ = 0.25), while greater participation and relevance led to greater empowerment in healthcare (γ = 0.76). In addition, greater critical consciousness led to greater participation and relevance (γ = 0.12). Finally, greater community capacity led to greater issues selection (γ = 0.56), which in turn led to greater participation and relevance (γ = 0.25). The study makes proposals for directions of health centers for workers and community networks. Confirmation of this model for worker empowerment suggests several directions to HCFW in relation to workers and community networks.

## 1. Introduction

With the increasing specialization and advancement of industrial technologies, businesses in South Korea have been downsized, resulting in more outsourced jobs. Against this backdrop, the number of small worksites in Korea has been on the rise. According to the Ministry of Employment and Labor (MoEL) [1], approximately 2.4 million Korean companies—that is, 98.2% of businesses across South Korea—hired fewer than 50 individuals in 2018. Overall, these companies employed 11.3 million individuals (59.4% of the total 19.1 million hires) in 2018. Although small, these businesses accounted for 78.3% of all industrial accidents at worksites, and employed 52.4% of all individuals who experienced an occupational disease in this same year [1]. These disproportionate numbers are largely due to the poor working conditions and older age of employees at small businesses [2]. In response, the MoEL and the Korean Occupational Safety and Health Agency created Health Centers for Workers (HCFW) in industrial complexes nationwide in order to give workers of small-scale companies access to health consultations, safety guidelines, and services for occupational disease prevention and basic health promotion. Currently, there are about 21 HCFW throughout South Korea. However, further strategies and programs that better consider local needs are necessary for these centers to play more active roles in healthcare. A number of national efforts to date have focused on increasing knowledge; improving individual attitudes via education and consultation; and improving organizational environments and abilities through changes in norms, capabilities, and atmosphere. Still, further local-level strategies are needed to manage unique local issues and mobilize community resources to ensure customized management of small workplaces nationwide [3].

Community organization development theory underscores the benefits of such local strategies, particularly focusing on identifying problems in the community and allocating resources to apply solutions. Common factors described as part of the theory are empowerment, community capacity, participation and relevance, critical consciousness, and issue selection [3,4].

While there have been several case studies focusing on topics such as building a network of community capacity [5,6], member participation [7,8,9], and empowerment [10,11], there is still relatively little research on critical consciousness and issue selection overall. Moreover, applying organizational theory to an HCFW requires an understanding of the impact of each factor and their interactions. The most important element in empowerment in healthcare is estimating community capacity, which involves identifying a problem, reinforcing leadership for tackling issues, building a community network, and facilitating the participation of residents. Participation and relevance mediates the relationship between critical consciousness and issue selection, as local residents ensure equal participation, share their rights, and develop a local agenda. These community organizational theories are needed for the healthcare of small-workplace workers at HCFW. The organization of HCFW aims at strengthening empowerment in healthcare for occupational disease prevention among local small-scale employers Empowerment is the process of securing the control of individual’s and community life in order to solve the problems of the community effectively, and it is the major goal of the development of community.

Previous studies have reported that the participation of local residents tends to increase with the development of community capacity [12,13], and employee participation mediates the relationship between community capacity and empowerment [14]. Hatcher et al. [15] also found an indirect relation between community capacity and empowerment in healthcare, which was driven by greater interaction and empathy between workers themselves and other workers. Furthermore, past research has similarly shown that issue selection is more than member participation [16], and different organizational systems display different levels of issue selection [17]. Previous studies have shown that community capacity has a significant impact on critical consciousness [18] and critical consciousness has a significant impact on participation in health initiatives [19,20].

Based on these previous findings, this study established an HCFW-based model of community capacity, critical consciousness, issue selection, and participation and relevance with a focus on employees working for small-scale employers (Figure 1). The results aim to improve health management ability and overall health conditions among individuals who are exposed to relatively poor working conditions in small businesses.

## 2. Materials and Methods

### 2.1. Study Setting

The area of B region is 53.45 km^2^, population 871,674, and the population density is 16,323 people/km^2^, which is very dense. The area had a total of 60,076 businesses and 291,822 workers in 2017. Among them, 99.1% (59,520) of businesses had less than 50 employees and those businesses employed 73.9% of 291,822 workers. Small businesses in the B region are concentrated in nine large apartment factories. The HCFW is located in a large apartment plant (Figure 2).

### 2.2. Data Collection and Ethical Consideration

Participants were employees of businesses with a workforce of less than 50 persons in the B region of South Korea, where an HCFW was in operation. For a structural equation model of 18 variables, a sample size of 180 participants or more was required [21]. The researcher sent a letter explaining the study to 50 employers in the manufacturing, transportation, and service industries, and 45 agreed to participate in. In September 2015, 550 questionnaires were sent to all workers in those companies. Questionnaires were given to employees with the consent of worksite managers, and participating employees were informed of the purpose and content of the survey. The number of responses received was 495. After filtering answer sheets that were missing values and deemed improper, we had 408 survey papers. This research secured more than 200 samples in consideration of the structural equation. The research was reviewed by the Institutional Review Board (IRB) of The Catholic University of Korea (MC15QISI0089).

### 2.3. Study Tool

All questionnaire was developed based on the existing literature on workplace and worker characteristics, empowerment in healthcare, community capacity, participation and relevance, critical consciousness, and issue selection and was verified by two academic experts in the field of health promotion, HCFW, and occupational health. All items in the questionnaire was rated on a 5-point Likert scale (1 = strongly disagree and 5 = strongly agree).

#### 2.3.1. Empowerment in Healthcare

Empowerment in healthcare refers to any action that contributes to self-efficacy, health information, and handling ability, and addresses a request for help [22]. The existing tool by Lee [23], the Change of Empowerment Scale, was modified to measure changes in employees’ health-management ability. The tool had 16 items in four subscales: self-efficacy (5 items), health information (6 items), handling ability (3 items), and help requests (2 items). Cronbach’s α for the tool was 0.93 in the original study by Lee (2000), and 0.90 in this study.

#### 2.3.2. Community Capacity

Community capacity refers to the ability of communities and HCFW to offer help, collaboration, and support for better health management. Three items were modified from a tool by Shin [24], the Relationship Capability Subscale, measuring community help, collaboration, and support, ultimately creating a scale with six items evaluating our community and HCFW. Cronbach’s α for this tool was 0.94 in this study.

#### 2.3.3. Participation and Relevance

Participation and relevance refer to employees’ intention to use an HCFW and address their own health issues, as well as their recognition of their own right to participate in their healthcare. To evaluate participation and relevance, the researchers developed a 15-item tool assessing workers’ motives for visiting the center (5 items), efforts to solve community health problems (6 items), and health rights (3 items). These items were adapted from the Change of Empowerment Scale by Lee [22]. Cronbach’s α for the tool was 0.74.

#### 2.3.4. Critical Consciousness

Critical consciousness refers to employees’ understanding of occupational disease, knowledge of the specific diseases related to their jobs, and efforts to avoid disease. To measure critical consciousness, four items from the Disease Sensitivity Scale [16] were adapted, which assess four areas of disease sensitivity: awareness of occupational disease, relevance to health management, relevance to work life, and relevance to self-regulation. Cronbach’s α in the study by Ahn [25] was 0.75, whereas it was 0.89 in this study. 

#### 2.3.5. Issue Selection

Issue selection refers to workers’ interest in personal and local health issues; to measure it, the researchers again collaborated with the experts to modify the Community Health Issues Subscale [24] to assess overall and community health issues. Cronbach’s α in the study by Shin [24] was 0.52, while in this study it was 0.76.

### 2.4. Data Analyses

The survey data were analyzed using PASW Statistics for Windows 18.0 and AMOS 18.0 (SPSS Inc., Chicago, IL, USA). General characteristics of employees and peculiarities of research variables are presented using descriptive statistics, and the reliability of the research tools was analyzed with Cronbach’s α. This study employed the item parceling method, which reduces indices by mixing diverse questions of measuring tools or calculating averaged values. To test the model of empowerment in healthcare of small-business workers and calculate the path coefficients between all variables, structural equation modeling was performed. To test the model fit, χ2 and root mean square error of approximation (RMSEA) were used as absolute fit indices, whereas the normed fit index (NFI) and comparative fit index (CFI) were used as incremental fit indices. To identify additional model paths and improve model fit, a modified model was tested and its model fit assessed (Figure 3). This study employed the modification indices derived from AMOS before modifying them [26]. In addition, squared multiple correlation (SMC) against endogenous variables was conducted between variables, and direct, indirect, and total effects among variables were examined.

## 3. Results

### 3.1. Demographic Characteristics

Those surveyed and their companies showed the following demographic characteristics. The average age of those surveyed was 46.40 ± 12.63 years; male employees were 241 (59.1%) in number and females 167 (40.9%). They had worked at their current work place for 7.57 ± 7.95 years on average. Three hundred thirty-two (81.4%) workers were permanently hired, while 76 (18.6%) were temporarily recruited. Those who had called in an absence due to illness numbered 29 (7.1%). In total, 291 (71.3%) participants were working in the manufacturing sector; 176 (43.1%) worked at companies that operated a safety committee (Table 1).

### 3.2. Descriptive Statistics and Normality Test

Measurement variables in structural equations assumingly follow normal distribution. To test this, this study examined kurtosis and skewness. The absolute values of skewness (3 or lower) and kurtosis (8 or less) showed normal distributions. In addition, there should be no multicollinearity issues among major variables. This was proven with the outcomes that all variance inflation factors were lower than 10, an indication that there was no such problem.

### 3.3. Test of the structural Model

#### 3.3.1. Test of the Community Organizational Model in Small Businesses’ Workers

To identify the additional model paths and improve model fit, the researchers tested a modified model and assessed its model fit (Table 2).

The results showed that variables with a significant direct influence on empowerment in healthcare were community capacity (γ = 0.25, *p* < 0.001) and participation and relevance (γ = 0.76, *p* < 0.001). The factors significantly influencing participation and relevance were community capacity (γ = 0.39, *p* < 0.001), critical consciousness (γ = 0.12, *p* = 0.021), and issue selection (γ = 0.25, *p* < 0.001). Community capacity did not have a significant influence on critical consciousness (γ = −0.08, *p* = 0.159), but significantly influenced issue selection (γ = 0.56, *p* < 0.001; Figure 2). The R2 values of the modified model were 0.82 for empowerment in healthcare, 0.33 for participation and relevance, and 0.31 for issue selection. Community capacity had a total effect of 0.52 on participation and relevance, including an indirect effect of 0.13 via critical consciousness and issue selection (Table 3).

#### 3.3.2. Direct, Indirect, and Total Effects

The total effect of community capacity on empowerment in healthcare was 0.64, including an indirect effect of 0.39 via critical consciousness, issue selection, and participation and relevance. Critical consciousness had a total effect of 0.09 on empowerment in healthcare, which solely comprised an indirect effect via participation and relevance; issue selection had a total effect of 0.89 on empowerment in healthcare via participation and relevance, with no direct effect (Table 4).

## 4. Discussion

HCFW were established for occupational disease prevention in local small-scale businesses in South Korea. This study sought to establish an HCFW-based model that explains the factors influencing empowerment in healthcare of employees at small-scale worksites. The models built in this study to test empowerment in healthcare revealed that community capacity centered on the HCFW had direct and indirect influences on participation and relevance in HCFW via critical consciousness, issue selection, and participation, and relevance directly affected empowerment in healthcare.

Community capacity also had strong direct and total effects on participation and relevance for the workers. Other studies have similarly reported that the participation of local residents tends to increase with the development of community capacity [12,13], and employee participation mediated the relationship between community capacity and empowerment [14,15]. Earlier studies showed that when a community views participation in occupational disease issues as negative or not widely accepted, residents tend to avoid managing the cause of occupational diseases [27]. In addition, participation tends to be determined by the participatory structure and circumstances of the community [23]. This study revealed that employees of small businesses exhibited different patterns of participation according to their recognition of HCFW-centered community capacity. Higher community capacity is believed to encourage participation for small-scale workers and renews support for managing occupational disease. In other words, when community resources are well organized around HCFW, employees tend to use them more. Participation can therefore be considered a critical factor for empowerment, particularly because it mediates between community capacity and empowerment [28].

The variable with the largest direct and overall effects on empowerment in healthcare was participation and relevance. This may be because more frequent use of healthcare services by employees leads to greater participation in local health issues [29], which in turn leads to a greater awareness of rights and empowerment [30]. This finding may be interpreted as follows: Employees increasingly feel the need to use HCFW when they develop an awareness of the risk of occupational diseases, prompting local communities to start dealing with those health issues more actively, thus strengthening employees’ understanding of their health rights.

Issue selection had the second-largest direct and total effects on participation and relevance. This is the same outcome as previous studies: Community capacity impacts issue selection [17] and issue selection influences participation and relevance [16]. Greater interest in issue selection triggers greater participation [31]. All these outcomes indicate that empowering employees in occupational health issues in HCFW and the community can be built through the following process: Community residents prepare themselves to help, collaborate, and support each other’s health, choose the necessary issues to tackle, and participate in efforts to address them.

Critical consciousness had a direct effect on participation and relevance, and this relationship was not mediated by any other variables. This is more evidence of the previous research in which critical consciousness impacted participation and relevance. [19,20]. Activities, including question-and-answer forums, are ways of impacting participation and relevance [15]. Therefore, HCFW and communities should implement strategies involving the participation of workers in discussions on occupational health in order to increase critical consciousness. In the meantime, this investigation was different from other studies in that it demonstrated that community capacity influenced critical consciousness [18]. Sheri et al. [32] found that community capacity and critical consciousness, among others, are prerequisites for empowerment. Yet, their research uncovered that the situation of South Africa unfolded the other way. This study reveals that the cultural factors of a community that are distinguishable from others were not impactful. However, this needs to be supported by follow-up studies.

Previous studies uncovered that the health management of employees working at small businesses required changes in managers [33], CEOs’ leadership, and organizations [34]. In an effort to better involve employers in the handling of occupational diseases, community opinion leaders might take part in grasping the current status and needs of the local community, fostering health leaders for employers, and running clubs that focus on improving employer participation. Increasing voluntary participation in the prevention of occupational diseases requires a transformation of the community’s health culture. The community’s health resources must be centered on HCFW, and the enhanced community capacity can improve networks between HCFW and workplaces. In South Korea, as part of an initiative to increase community capacity, HCFW have begun to commission local opinion leaders to form community management committees, periodically operate steering committees, and strengthen workplaces and agreements to provide community-based integrated services.

Above all, efforts should focus on improving community capacity first, followed by promoting critical consciousness and issue selection. As critical consciousness develops during the discussion and sharing of issues, tools to increase awareness of occupational diseases must be developed at the local level. HCFW should address diverse issues for employee issue selection. To promote critical consciousness in relation to occupational diseases, HCFW have held campaigns and training sessions for small-business employers. HCFW have also strengthened public relations efforts to help improve issue selection, such as by issuing press releases related to the opening of HCFW, service content, and staff awards. These press releases are effective, as most workers rely on actually visiting the HCFW to obtain information about them.

Other countries followed suit. For employees’ health management, Japan established regional industrial health center. The Austrian government came up with programs that support industrial safety health management. Finland has put municipal health care centers in place. The UK is running pilot programs: National Institutes of Health (NHS) Plus program and Fit-For-Work [35]. Those policies and programs show trends that owners of small businesses that lack monetary resources are making their government and local communities work for the health management of employees at small-sized workplaces. This study contributes to the idea of suggesting models that local communities should select at the time of implementing the health management of workers at small businesses.

As this was a cross-sectional survey, causal relationships between the variables could not be confirmed. Nevertheless, this study is significant because it is among the first studies researching health promotion centers for employees working at small-scale businesses in South Korea. Furthermore, it overcomes the limits of existing research by applying community organization theory and demonstrating effects via structural models. This investigation also comes with limitations, as previous studies on health management of small businesses’ employees are scarce. In the future, research that can reduce occupational diseases and work-related illnesses for those working under poor and aggravating environments of small businesses should be carried out.

## 5. Conclusions

This study evaluated a health empowerment model designed for employees working at small-scale businesses to improve their health. It surveyed individuals in relation to HCFW, and tested structural equation models based on community organizational development theory. The results indicated that empowerment in healthcare can be achieved through devising a community cooperation system based on HCFW that seeks to raise community capacity, after which community health issues should be identified and critical consciousness regarding these issues improved through discussion. These efforts, in turn, might lead to improvements in participation and relevance, and subsequently, empowerment.

Given these results, the researchers argue that there was significant progress in the elements of community organizational development theory that centered on an HCFW and in the model designed to lead to empowerment. For future direction of health promotion centers and local community networks, the researchers offer the following proposals: First, for a stronger community capacity, networks between HCFW and worksites should be improved. Second, as critical consciousness occurs in the course of discussions about and sharing of issues, debates and other tools for awareness of occupational diseases should be developed at local levels. Third, limited staff sizes put small businesses under greater community influence. In order to increase voluntary participation in prevention of occupational diseases, a community’s health culture should be transformed.

## Figures and Tables

**Figure 1 ijerph-17-02444-f001:**
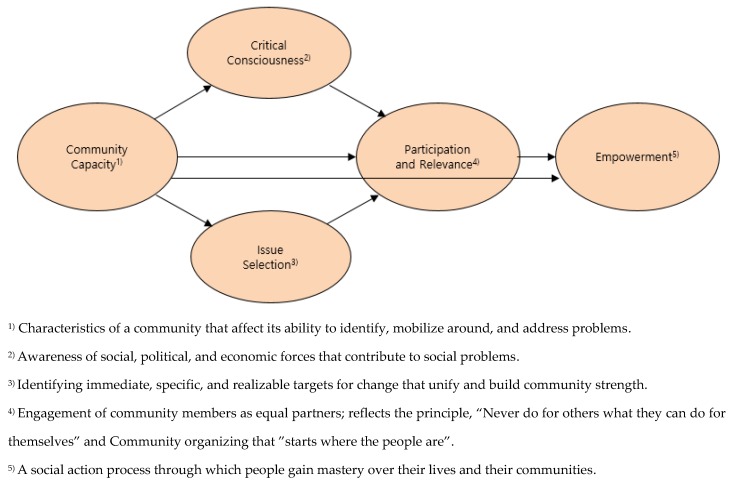
Conceptual framework.

**Figure 2 ijerph-17-02444-f002:**
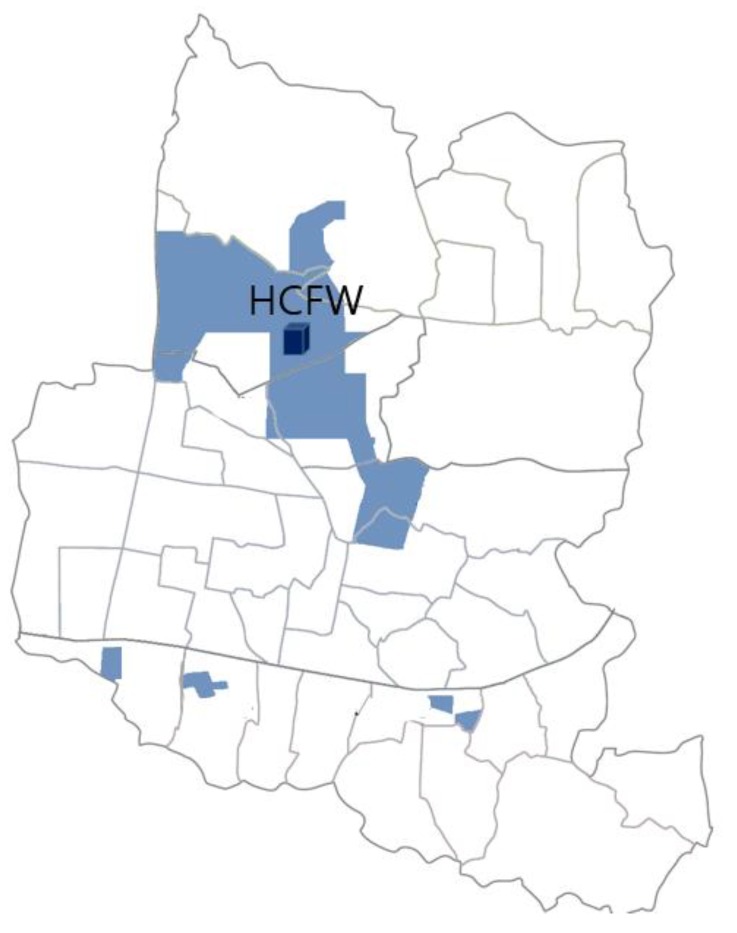
Small businesses and health centers for workers (HCFW) locations in B area.

**Figure 3 ijerph-17-02444-f003:**
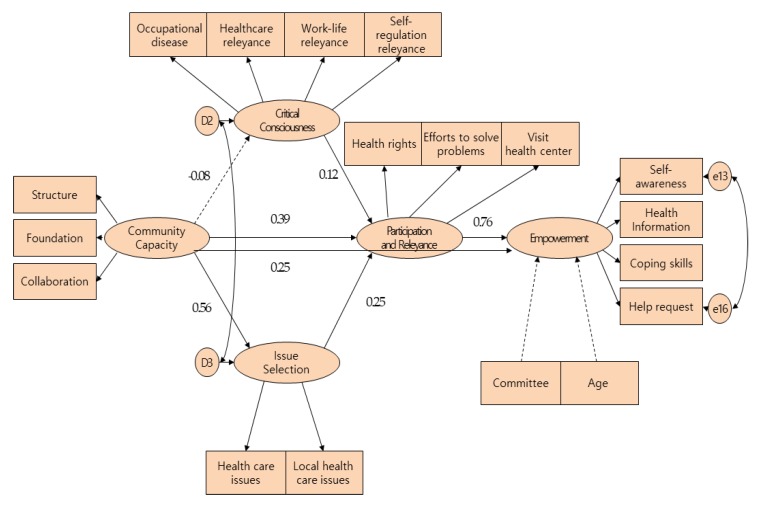
Path of modified model. **χ^2^ = 382.91** (df = 126), *p* ≤ 0.001 RMSEA = 0.07, NFI = 0.92, CFI = 0.95.

**Table 1 ijerph-17-02444-t001:** Demographic characteristics of study participants (*n* = 408).

Classification	Category	*N*	%
Age group (years)	˂30	52	12.7
	30–39	76	18.6
	40–49	112	27.5
	≥ 50	168	41.2
Mean ± SD		46.40 ± 12.63	
Gender	Male	241	59.1
	Female	167	40.9
Years of career experience	˂ 5	220	53.9
	5–9	79	19.4
	≥10	109	26.7
Mean ± SD		7.57 ± 7.95	
Employment type	Permanent	332	81.4
	Temporary	76	18.6
Absence due to illness	Yes	29	7.1
	No	379	92.9
Business type	Manufacturing	291	71.3
	Non-manufacturing	117	28.7
Safety and health committee	Yes	176	43.1
	No	232	56.9

**Table 2 ijerph-17-02444-t002:** Comparison of fit of the modified and hypothetical models.

Model	CMIN			RMSEA	NFI	CFI
	χ^2^	df	*p*			
Acceptance criteria			<0.05	≤0.10	≥0.90	≥0.90
Hypothetical model	414.35	128	<0.001	0.07	0.92	0.94
Modified model	382.91	126	<0.001	0.07	0.92	0.95

*Note.* RMSEA = root mean square error of approximation; NFI = normed fit index; CFI = comparative fit index.

**Table 3 ijerph-17-02444-t003:** Standardized path coefficients of the modified model of the community organizational model of those working at small businesses in Korea.

Endogenous Variables	Exogenous Variables	Standardized Estimation	Standard Error	Critical Ratio	*p*	SMC
Consciousness	Capacity	–0.08	0.06	−1.41	0.159	0.01
Issue selection	Capacity	0.56	0.05	8.75	<0.001	0.32
Participation and Relevance	Capacity	0.39	0.03	5.82	<0.001	0.34
	Consciousness	0.12	0.02	2.46	0.014	
	Issue selection	0.25	0.04	3.69	<0.001	
Empowerment	Capacity	0.25	0.03	6.75	<0.001	0.82
	Participation and Relevance	0.76	0.12	11.56	<0.001	

*Note.* SMC = squared multiple correlation.

**Table 4 ijerph-17-02444-t004:** Direct effect, indirect effect, and total effect.

Variables	Participation and Relevance			Empowerment		
	Direct	Indirect	Total	Direct	Indirect	Total
Community Capacity	0.388	0.132	0.520	0.250	0.393	0.644
Consciousness	0.118	-	0.118	-	0.089	0.089
Issue Selection	0.250	-	0.250	-	0.289	0.289
Participation and Relevance	-	-	-	0.756	-	0.756

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
