# Peer review of "Models for Developing Community Organizations to Reinforce Health Management in Small Businesses"

_ijerph, 2020, doi:10.3390/ijerph17072444_

Round 1
Reviewer 1 Report
It is an original study with interesting results. The study highlights the importance of community capacity and participation and relevance for empowermentli in healthcare in Health Centers for Workers in Korea. My suggestion is to to deepen into the importance of healthcare and employers' consciousness and nin colusions, highlight future research lines. Also, the map shown on page 3 is not quite clear. Maybe the title could be modified, since the research not only reflects the proposals for the activation of Health Centers, the main variables could be highlighted in the title.
Author Response
I would like to extend my deep appreciation to you for review. I thankfully found this round of editing a good chance of improving the paper.
- My suggestion is to to deepen into the importance of healthcare and employers' consciousness and nin colusions, highlight future research lines.-
“Other countries followed suit. For employees’ health management, Japan established regional industrial health center. The Austrian government came up with programs that support industrial safety health management. Finlad has put municipal health care centers in place. The UK is running pilot programs: NHS Plus program and Fit-For-Work. Those policies and programs show trends that lack of monetary resources of owners of small businesses are making their government and local communities work for the health management of employees at small-sized workplaces. This study contributes to the idea of suggesting models that local communities should select at the time of implementing the health management of workers at small businesses.
As this was a cross-sectional survey, causal relationships between the variables could not be confirmed. Nevertheless, this study is significant because it is among the first studies that research health promotion centers for employees working at small-scale businesses in South Korea. Furthermore, it overcomes the limits of existing research by applying community organization theory and demonstrating effects via structural models. . This investigation also comes with limitations as previous studies on health management of small businesses’ employees are in scarcity. In the coming days, such research that can reduce occupational diseases and work-related illnesses for those working under poor and aggravating environments of small businesses should be carried out.” - the map shown on page 3 is not quite clear.
- The map was changed more cleanly. - Maybe the title could be modified, since the research not only reflects the proposals for the activation of Health Centers, the main variables could be highlighted in the title.
- Models of Developing Community Organizations for Reinforcing of Health Management at Small Businesses

Reviewer 2 Report
Introduction
As for the first section that addresses the present work, certain deficiencies in the content are observed as a theoretical basis to support the exposed results.
- In the first place, it is observed how the quotations exposed in this section are outdated, therefore, it is suggested to update them based on more updated content of the last two years. In addition, after analyzing the results presented, there is a low theoretical base by the authors, with only 11 citations.
- In the second place, each of the variables treated in its study will be addressed in this section (Empowerment, community capacity, participation and relevance, problem selection, and critical awareness), as well as the relationship between them. In this way, it would be possible to contrast their results with those of other previous investigations, since currently, with the above introduction it would be unfeasible to establish this comparison. Therefore, the theoretical argument of previous studies on community capacity and its direct effect with participation and relevance, as well as indirect mediation of participation and relevance, and critical awareness of capacity can be raised community and empowerment. In addition, to address the relationship between participation and relevance and community capacity, problem selection, and critical awareness.
- Thus, you are advised to address empowerment and its direct relationship with participation and relevance, and community capacity, as well as the Issue selection and their relationship with empowerment through participation and relevance. Finally, you should also talk about critical consciousness and its association with empowerment through participation and relevance.
Method
- Regarding the method, you should clarify the total number of the sample, this issue is currently not clear.
Results
- Regarding the results, in the first section where it reflects the demographic characteristics, it should indicate the standard deviation of the age data in the text and the years of career experience.
- Regarding Table 2, you must add a new column by adding the X2 / (gl).
Discussion
- The discussion is a comparison between their results and those found by other cited authors, previously in the introduction. Therefore, it is observed that all the studies that appear in the discussion do not appear in the introduction and should be present in the introduction.
- You must include in this section future lines of research.
- As a limitation, I could add, as mentioned above, that there are a small number of studies that relate the variables treated in this paper.
Conclusions
- The information that appears on lines 275, 276, 277 and 278 appears in duplicate since it has been mentioned previously on lines 230, 231, 232, 233, and 234. Therefore, you must remove it from the conclusions section.
“As this was a cross-sectional survey, causal relationships between the variables could not be 275 confirmed. Nevertheless, this study is significant because it is among the first studies that research 276 health promotion centers for employees working at small-scale businesses in South Korea. 277 Furthermore, it overcomes the limits of existing research by applying community organization theory 278 and demonstrating effects via structural models.”
Author Response
I would like to extend my deep appreciation to you for review. I thankfully found this round of editing a good chance of improving the paper.
Please find my responses in the attachment.

Round 2
Reviewer 2 Report
After the changes made by the authors, a substantial improvement in the content is observed. Therefore, it is accepted.